# Ultrasonography Assessment Based on Muscle Thickness and Echo Intensity in Post-Polio Patients

**DOI:** 10.3390/diagnostics12112743

**Published:** 2022-11-09

**Authors:** Álvaro Mateos-Angulo, José Andrés Salazar-Agulló, Cristina Roldán-Jiménez, Manuel Trinidad-Fernández, Antonio Ignacio Cuesta-Vargas

**Affiliations:** 1Department of Physiotherapy, Universidad de Málaga, 29071 Málaga, Spain; 2Grupo Clinimetría F-14, Instituto de Investigación Biomédica de Málaga—IBIMA Plataforma BIONAND, 29590 Málaga, Spain; 3Centro de Salud El Palo, 29018 Málaga, Spain; 4School of Clinical Sciences of the Faculty of Health at the Queensland, University of Technology, Brisbane 4072, Australia

**Keywords:** neuromuscular diseases, post-poliomyelitis syndrome, ultrasonography, muscle strength, postviral fatigue syndrome

## Abstract

There is no specific designed diagnostic test for post-poliomyelitis syndrome. The most important symptoms of this syndrome are new loss of muscle strength and more fatigue. Previous studies have investigated muscle ultrasound parameters to distinguish neuromuscular disease patients from healthy controls. The aim of this study was to investigate if muscle thickness and echo intensity measured by ultrasound can discriminate post-poliomyelitis syndrome patients from healthy controls. A total of 29 post-polio patients and 27 healthy controls participated in this cross-sectional study. Anthropometric measures, muscle thickness, echo intensity using B-mode ultrasound in rectus femoris and biceps brachii muscles, and muscle strength test data were collected. Muscle thickness in rectus femoris was significantly lower in post-poliomyelitis patients than in healthy controls, but not in biceps brachii. Echo intensity in rectus femoris and biceps brachii was higher in post-poliomyelitis syndrome patients than in healthy controls. Correlations were found between muscle thickness and strength in the upper and lower limbs. The results of the present study showed that muscle thickness in rectus femoris and echo intensity in rectus femoris and biceps brachii can discriminate post-poliomyelitis syndrome patients from healthy controls. A better assessment is possible because it can observe differences and relevant parameters in this clinical population.

## 1. Introduction

Patients after bacterial and viral infections often share a cluster of symptoms in the long term, such as fatigue and muscle weakness. For example, fatigue is the general symptom with the highest prevalence in long COVID-19, affecting between 58% and 95.9% of patients [1,2], and muscle weakness is reported in up to 63% of COVID-19 patients at 6 months post-discharge from the hospital [3]. These symptoms are also shared by patients who suffered from poliomyelitis, one of the most acutely debilitating infections that affected millions of people in the 1940s and 1950s [4]. Specifically, these patients may experience new-onset symptoms such as fatigue and muscle weakness. When this onset occurs later in life and after an interval of neurological recovery and stability, it is defined as post-poliomyelitis syndrome (PPS) [5,6].

The pathogenesis of PPS is controversial and not completely understood. Currently, PPS diagnosis consists of inclusion of diagnostic criteria, but a specific diagnostic test has not been designed [7]. Key diagnostic criteria for PPS include prior paralytic poliomyelitis followed by a time of convalescence and stability, the new loss of muscle strength or fatigue for a minimum of a year, and exclusion of other neurologic, medical, or orthopedic problems [7,8]. Previous studies have investigated the possible diagnostic evaluations for PPS syndrome, including creatine kinase levels, muscle biopsy, and electrodiagnosis methods [9,10,11,12], but there are no current neurophysiological findings that would differentiate between stable post-polio patients with or without PPS. For this reason, the investigation into new methods for diagnosing PPS patients is required.

PPS disease is commonly evaluated with functional tests and muscle strength measurements. In this context, imaging studies, such as ultrasonography, may be indicated in PPS [7]. An ultrasound is a low cost and safe tool that has been previously used to evaluate neuromuscular diseases [13]. Ultrasound variables, such as muscle thickness and echo-intensity, have shown a relationship between function and strength in several clinical populations [14,15]. A prior study investigated muscle architecture and muscle quality in patients with post-polio syndrome, showing that muscle ultrasound can identify differences in muscle thickness and echo intensity in lower limb muscles comparing healthy controls and PPS patients [16]. The purpose of the present study was to investigate if static and dynamic muscle ultrasound parameters in the upper and lower limb can discriminate PPS patients from healthy controls. Furthermore, the difference in the lower limbs more and less affected was investigated in PPS patients. A tertiary objective was to study the relationships between muscle thickness, echo intensity, and muscle strength in PPS patients.

## 2. Materials and Methods

### 2.1. Experimental Subjects

A total of 29 adults (19 women and 10 men) diagnosed with PPS were recruited between April and July 2018 from a local association of polio and post-polio patients in Málaga, Spain; 27 healthy age-matched controls participated in the study. The inclusion criteria were: polio survivors with new symptoms after 15 years stable; the progressive appearance of muscle weakness, atrophy, pain, sensory disturbances, and fatigue; persistence of new symptoms for 1 year; and no other neurological, medical, or orthopedic diseases that generate the symptoms. Patients were excluded if they had a surgical intervention during the last year or any neurologic and trauma additional condition that was impossible to carry out the tests.

The study complied with the principles laid out in the Declaration of Helsinki and was approved by the Ethics Committee of the University of Malaga CEUMA 52-2018-H. Patient assessment and identification of cases and controls were performed independently of the ultrasound assessment. The ultrasound evaluator was blind to the rest of the assessment. Participants were given a detailed description of the study and signed written informed consent for inclusion in the study.

### 2.2. Anthropometric Measurements

Height and weight variables were collected following the guidelines of The International Society for the Advancement of Kinanthropometry (ISAK) [17]. Additionally, fat average and lean mass were assessed with a bioimpedance scale (Omron Body Composition Monitor BF511, Osaka, Japan).

### 2.3. Muscle Strength

Knee extensor maximum voluntary contraction (MVC) muscle strength was evaluated by a digital manual dynamometer (POWERTRACK^®^ Jtech Medical. Midvale, UT, USA). An interclass correlation coefficient (ICC) ranging from 0.72 to 0.85 has been previously demonstrated for this dynamometer [18]. The dynamometer shows the maximum force in Newtons achieved in each test. Each participant performed three repetitions of the test. The rest time was two minutes between repetitions. The maximum score in Newtons of all three tests was taken.

### 2.4. Muscle Thickness and Echo Intensity

An ultrasound device (MyLab One, Esaote, Genoa, Italy) with a linear array transducer (SL3323, Esaote, Genoa, Italy) with a variable frequency of up to 22 MHz was used for the muscle B-mode ultrasound examination. The US parameters were 70% gain, and the default settings for the grayscale map and the dynamic range were selected. The exported image had a size of 800 × 652 px with 96 dpi. The examiner had more than 4 years of experience in musculoskeletal ultrasound in a clinical context.

Two transversal static B-mode images in repose and MVC on each side of the patient were acquired. To achieve the best possible muscle image quality, the minimum pressure was applied on the skin using a lot of ultrasound gel, and superficial and deep aponeuroses representation was optimized. The same operator carried out a single measurement session to acquire the images (see Figure 1).

The following upper and lower limb muscles were examined: biceps brachii (BB) and rectus femoris (RF). The transducer was placed following the instructions of previous studies to find the anatomical location of the largest diameter of the examined muscles [19,20]. BB was measured at two-thirds of the distance from the acromion to the antecubital crease, and the RF was measured at a mean distance from the anterior-superior iliac spine to the superior border of the patella.

Ultrasound images were analyzed offline with ImageJ (ImageJ, NIH, Bethesda, MD, USA) software. A known distance showed that the ultrasound display was used to calibrate the rule tool of ImageJ and calculate the muscle thickness. Both images collected from the participants were measured and averaged (Figure 2). The distance between deep and superficial aponeuroses was used to calculate muscle thickness. Echo intensity is the average result of a histogram of the 8-bit grayscale and was calculated by the grayscale analysis function in ImageJ software of a region of interest previously selected, which included the possible muscle area avoiding the muscle fascia. There was no unit for this analysis.

### 2.5. Statistical Analysis

A database was created with the evaluation data collected to analyze the results. Descriptive statistics were performed with measures of central tendency and dispersion of variables. The difference in means from the case and control sample was compared using A Student’s *t*-test. Two analyses were performed depending on the muscle: (a) due to the functional affection being predominantly in the lower limbs, it was compared between the most affected lower limb in PPS and the dominant side limb in the control group; (b) dominant side of the upper limb was compared between PPS and the control group. A Student´s *t*-test was also used to compare the limbs depending on the affectation in PPS, excluding patients where the affectation was absolute in both limbs. Univariate correlations with strength and ultrasound variables were tested with the Pearson correlation coefficient. Significance was set at *p* < 0.05. All statistical analyses were performed with SPSS 21 (SPSS Inc., Chicago, IL, USA) software package.

## 3. Results

Table 1 summarizes the anthropometric characteristics of cases and controls. About 51.7% of the case sample had lower limb affectation on the right side, 37.9% on the left side, and 10.3% on both sides. It was not possible to measure the ultrasound variables of the most affected lower limbs in one patient because they did not allow the removal of the orthosis they had, which made it impossible to collect data. Table 2 shows the mean values of echo intensity, muscle thickness, and strength. RF muscle thickness in repose and MVC was significantly lower (*p* < 0.001) in PPS patients than in healthy controls but not in BB (see Table 2). Figure 3 represented a demonstration of this difference. The echo intensity in RF and BB was higher in PPS patients than in healthy controls (*p* = 0.001–0.01) (see Table 2). Knee extensor strength was lower in PPS than in healthy controls (*p* < 0.001) (see Table 2). Table 3 presented significant differences in muscle thickness and echo intensity between the most affected lower limb and the other one (*p* = 0.001–0.007). Strength showed significant correlations with muscle thickness (r = 0.51–0.72) and echo intensity in RF (r = −0.38) (see Table 4). All the correlation graphs were presented in Figure 4.

## 4. Discussion

The present study analyzed the differences between PPS patients and healthy controls using muscle thickness and echo intensity ultrasonography parameters. This is the first study to investigate dynamic ultrasound assessment (muscle thickness in MVC) in PPS. As a result, muscle echo intensity was the US outcome that better discriminated between PPS patients and healthy controls. While the echo intensity showed differences between healthy controls and PPS patients in RF and BB muscles, muscle thickness showed only differences in RF muscle (see Table 2).

Ultrasound is a valid tool to evaluate specific muscles [21] and is a clinically proven safe tool for the diagnosis of neuromuscular disorders [13]. Previous studies have used ultrasonography to evaluate disease severity and progression in amyotrophic lateral sclerosis [22,23], such as Duchenne muscular dystrophy [24], glycogen storage disease [25], or spinal muscular atrophy [26]. Additionally, a previous study investigated muscle ultrasound as an evaluation tool to discriminate PPS patients from healthy controls in the vastus lateralis muscle [16]. However, this is the first study to evaluate muscle mass by muscle thickness and muscle quality by echo intensity in the upper and lower limbs by ultrasound in PPS patients compared to healthy controls.

Regarding echo intensity, the results of the present study showed that PPS patients had a higher echo intensity in RF muscle (*p* < 0.001) and BB muscle (*p* = 0.01) compared to healthy controls. These results are similar to the study by Bickerstaffe et al. (2015), which showed that PPS patients had a higher echo intensity (*p* < 0.001) of the vastus lateralis muscle than healthy controls [16]. However, Bickerstaffe et al. (2015) performed no dynamic ultrasound assessments and indicated that future research was needed [16]. It should be noted that an increase in the echo intensity indicate increased infiltration of fat or fibrous tissue [27] as found in other clinical populations, such as amyotrophic lateral sclerosis [28].

According to the muscle thickness, the present study found that RF muscle thickness in MVC was significantly lower in PPS patients than in healthy controls (*p* < 0.001). Findings in this outcome also concur with Bickerstaffe et al. (2015), who found PPS patients had a lower muscle thickness (*p* = 0.013) than healthy controls [16]. On the other hand, we found a positive correlation between RF muscle thickness and strength (r = 0.58, *p* < 0.001), and a negative correlation between RF muscle echo intensity and strength (r = −0.38, *p* < 0.05). The results of Bickerstaffe et al. showed similar results in the vastus lateralis muscle thickness (r = 0.480, *p* = 0.001) and the vastus lateralis muscle’s echo intensity (−0.463, *p* = 0.001) [14]. The change in this outcome in MVC has also been found to distinguish between pathological and normal muscles in other clinical populations, such as patients with muscular myositis [29].

Along with an increase in echo intensity, a decrease in muscle thickness indicates muscle atrophy [30,31,32]. However, in addition to muscle atrophy, PPS patients present other symptoms such as myalgia, arthralgia, dysphagia, and fatigue, as a consequence of neurological deficits. This neurological affectation has been evidenced by motor unit loss and denervation measured by electromyography, or changes in muscle histology [33]. Neurological affectations are also manifested in other post-viral infections, such as COVID-19, although given its novelty, research mainly reports neurological symptoms [34]. Future research should include both ultrasound and electromyographic measures in PPS to delve into the neurological affectation, which may help clinicians to better understand the sequelae of long COVID-19 patients.

This study presents several limitations. On one hand, studies in patients diagnosed with PPS imply a limitation itself, as there is not any laboratory test or objective outcome to differentiate between patients who suffered from poliomyelitis with or without PPS, and the diagnosis of PPS is based on symptom criteria. On the other hand, the present study analyzed differences between patients with PPS and healthy controls. Future studies should compare ultrasound parameters in polio patients with and without PPS. Lastly, future research should include the present US to determine their responsiveness to therapeutic interventions as well as to objectively assess the progression of the illness and muscular decline over time.

## 5. Conclusions

In conclusion, muscle thickness and muscle quality measured by echo intensity can discriminate between PPS patients and healthy controls. Muscle thickness in RF was lower in PPS patients than in healthy patients, but not in BB. PPS patients showed worse muscle quality (higher echo intensity) than healthy patients in BB and RF muscles. Muscle ultrasound parameters can be used to evaluate severity in PPS patients.

## Figures and Tables

**Figure 1 diagnostics-12-02743-f001:**
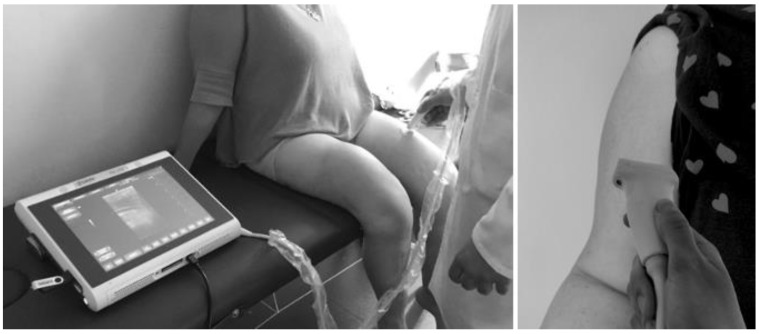
Assessment of muscle parameters in rectus femoris and biceps braquii by B-mode ultrasound.

**Figure 2 diagnostics-12-02743-f002:**
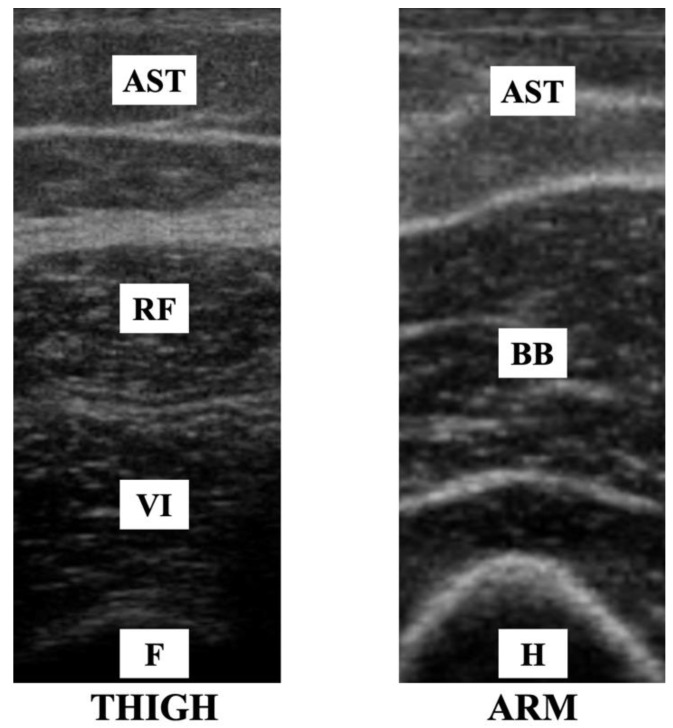
Muscles and adjacent structures on ultrasound images depending on the transductor placement. AST, adiposity subcutaneous fat; RF, rectus femoris; VI, vastus intermedius; BB, biceps braquii; F, femur; H, humerus.

**Figure 3 diagnostics-12-02743-f003:**
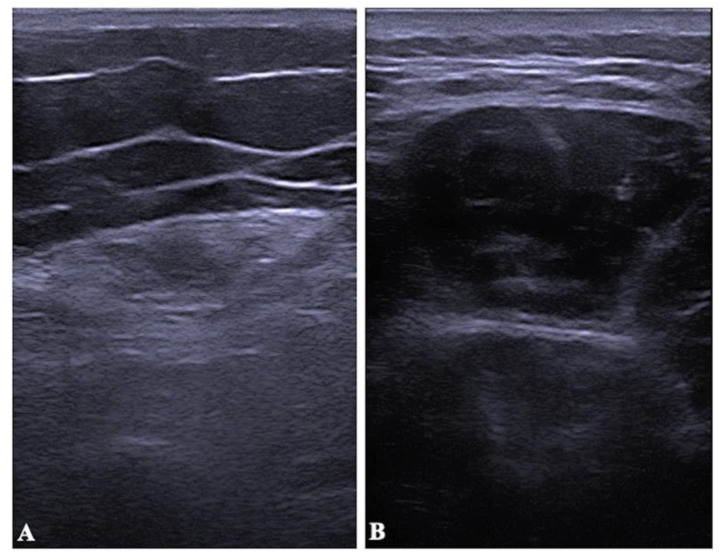
Ultrasound images of the rectus femoris of patients with PPS (**A**) and the healthy controls (**B**).

**Figure 4 diagnostics-12-02743-f004:**
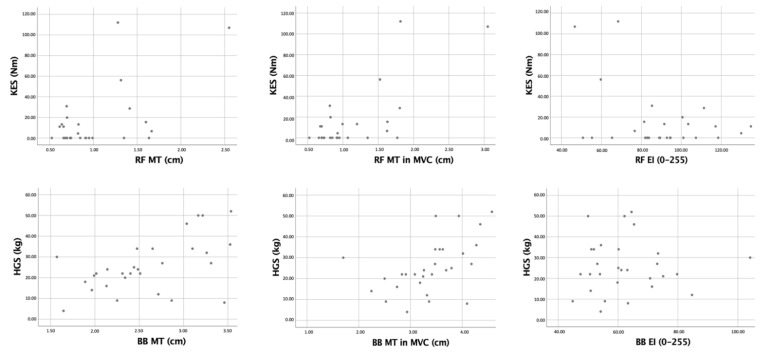
Graphs from the correlation analysis between the ultrasound outcomes and strength.

**Table 1 diagnostics-12-02743-t001:** Anthropometric characteristics.

	Cases	Controls	*p*
Age (years)	56.96 (3.44)	53.51 (12.99)	0.19
Height (m)	1.61 (0.13)	1.64 (0.07)	0.35
Weight (kg)	71.25 (12.91)	74.35 (12.16)	0.36
BMI (kg/m^2^)	27.20 (3.90)	27.60 (4.40)	0.72
Fat average (%)	39.16 (8.30)	36.01 (8.54)	0.19
Lean mass (kg)	25.30 (4.50)	27.56 (4.05)	0.07

Data is shown in Mean (Standard Deviation).

**Table 2 diagnostics-12-02743-t002:** Muscle ultrasound parameters and strength.

	Cases Mean (SD)	Controls Mean (SD)	*p*-Value	*t*-Student
BB MT (cm)	2.59 (0.56)	2.49 (0.42)	0.70	0.48
BB MT in MVC (cm)	3.37 (0.66)	3.30 (0.45)	0.42	0.67
RF MT (cm)	0.99 (0.45)	1.96 (0.37)	<0.001	−8.37
RF MT in MVC (cm)	1.11 (0.54)	2.28 (0.40)	<0.001	−8.80
BB EI (0–255)	62.38 (12.88)	52.03 (15.80)	0.01	2.62
RF EI (0–255)	89.16 (22.31)	54.68 (13.30)	<0.001	6.72
HGS (kg)	25.65 (12.80)	29.57 (8.41)	0.19	−1.32
KES (Nm)	15.55 (29.06)	154.40 (42.84)	<0.001	−13.90

SD: standard deviation; BB: biceps brachii; RF: rectus femoris; MT: muscle thickness; EI: echo intensity; MVC: maximum voluntary contraction; HGS: hand grip strength; KES: knee extensors.

**Table 3 diagnostics-12-02743-t003:** Rectus femoris analysis depending on the affected side in PPS patients.

	Most Affected (SD)	Less Affected (SD)	*p*-Value	*t*-Student
RF MT (cm)	0.98 (0.47)	1.27 (0.43)	0.004	−3.19
RF MT in MVC (cm)	1.11 (0.56)	1.45 (0.48)	0.007	−2.96
RF EI (0–255)	93.47 (20.59)	76.79 (24.27)	<0.001	6.11

SD: standard deviation; RF: rectus femoris; MT: muscle thickness; EI: echo intensity; MVC: maximum voluntary contraction.

**Table 4 diagnostics-12-02743-t004:** Correlations of ultrasound parameters with strength in PPS patients.

Variable	Pearson Coefficient
BB MT (cm)	0.53 *
BB MT in MVC (cm)	0.51 *
RF MT (cm)	0.58 **
RF MT in MVC (cm)	0.72 **
BB EI (0–255)	0.15
RF EI (0–255)	−0.38 *

* *p* < 0.01; ** *p* < 0.001; BB: biceps brachii; RF: rectus femoris; MT: muscle thickness; EI: echo intensity; MVC: maximum voluntary contraction.

## Data Availability

The datasets used and/or analyzed during the study are available from the corresponding author on reasonable request.

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
