# Peer review of "Ultrasonography Assessment Based on Muscle Thickness and Echo Intensity in Post-Polio Patients"

_diagnostics, 2022, doi:10.3390/diagnostics12112743_

Round 1

Reviewer 1 Report

Dear Authors,

The objective of this study was to investigate if muscle thickness and echo intensity measured by ultrasound can discriminate post-poliomyelitis syndrome patients from healthy controls. However, after careful reading I find that it is interesting and have same novelty that can be use in clinical practise. I have no major point to be considered concerning this paper. 

Best regards

Author Response

We appreciate the comments made on our work. Thanks for the recommendation.

Reviewer 2 Report

This study examined the usefulness of ultrasonography in the discrimination between post-polio syndrome and normal control.

They measured the muscle thicknesses and echo intensity of the biceps brachii and rectus femoris. The authors stated the aim of this study as follows, "There are no current neurophysiological findings that would differentiate between stable post-polio patients with or without PPS. For this reason, the investigation into new methods for diagnosing PPS patients is required." However, they also mentioned that "the present study analyzed differences between patients with PPS and healthy controls. Future studies should compare ultrasound parameters in polio patients with and without PPS." as a limitation. This gap seems to be critical. The reviewer thinks their study is still in its preliminary stage,

Moreover, the results might be standard features in various muscular disorders.

The reviewer would like to suggest comparing affected-side and unaffected-side muscles within individuals.

The reviewer wonders whether there are differences between cases and controls regarding anthropometric characteristics in Table1.

Please clarify the location of the company in lines 76-77.

The schematic illustrated will be helpful for readers who are not familiar with muscle sonography to orient the images in Figure 2.

Please reconsider the order of rows in Table 3. In the current manuscript, the order is "BB MT,  BB MT MVC,   RF MT,  RF MVC,   RF EI,  BB EI."  What does the order mean?

What does the term "de US outcome" mean in line 152? Spanish?

There is a typographical error in line 175 "found that found that"

The section on results consists of only legends for Tables. The authors should move the descriptions in lines 165-185 to the Results section.

Author Response

Thank you very much for reviewing our manuscript and for providing us with detailed suggestions on how to improve it. We have addressed all the comments expressed from reviewers and hope that the explanations and revisions are satisfactory. You can find clarifications for each comment. Changes were highlighted in yellow in the manuscript, please see the attachment.

  • They measured the muscle thicknesses and echo intensity of the biceps brachii and rectus femoris. The authors stated the aim of this study as follows, "There are no current neurophysiological findings that would differentiate between stable post-polio patients with or without PPS. For this reason, the investigation into new methods for diagnosing PPS patients is required." However, they also mentioned that "the present study analyzed differences between patients with PPS and healthy controls. Future studies should compare ultrasound parameters in polio patients with and without PPS." as a limitation. This gap seems to be critical. The reviewer thinks their study is still in its preliminary stage.

Thanks for the comments. We wanted to mean with this comment that this study is the first step in finding useful ultrasound variables in this type of patient. Thanks to these results, we have a preliminary idea of which variables may be optimal for use. From now on, other types of studies are necessary to consolidate and put the spotlight on these variables, such as reference values studies or responsiveness analysis. We cannot forget that Postpolio syndrome is a different syndrome from polio syndrome plus the effects of age. Another important future step is to look for differences between these two groups so that patients with post-polio syndrome have an excellent evaluation because they deserve it.

  • Moreover, the results might be standard features in various muscular disorders.

Thanks for the suggestion. Following your suggestion and other comments, we have presented a more complete analysis of the study presented for a better understanding by the reader that makes the work more relevant. We have made the comparison between cases and controls first with the most affected lower limb of the cases and the predominant limb of the healthy ones, which is the closest thing to a normal neuromuscular structure. Because the involvement in this syndrome is centered on the upper limbs, the biceps brachii was compared with the predominant side of both groups. Then, we made an intra-subject comparison between the most affected side and the less affected side of the lower limb to see the importance of this analysis in the ultrasound variables. All this information has been included in the Methods section. With this approach, we think that we have substantially improved the paper thanks to your thoughts.

  • The reviewer would like to suggest comparing affected-side and unaffected-side muscles within individuals.

DONE. Thanks for the recommendation. We included a comparison between affected-side and unaffected-side in the lower limbs where the affectation appears.

  • The reviewer wonders whether there are differences between cases and controls regarding anthropometric characteristics in Table1.

DONE. Thanks for the recommendation. We included the differences about anthropometric characteristics in Table 1.

  • Please clarify the location of the company in lines 76-77.

DONE. Thanks for the remark. We clarified the location of the Omron company.

  • The schematic illustrated will be helpful for readers who are not familiar with muscle sonography to orient the images in Figure 2.

DONE. Thanks for the recommendation. We include a new figure with the structures in both cross-sectional images.

  • Please reconsider the order of rows in Table 3. In the current manuscript, the order is "BB MT,  BB MT MVC,   RF MT,  RF MVC,   RF EI,  BB EI."  What does the order mean?

DONE. Thanks for the advice. We ordered Table 2 and 3 in the same way.

  • What does the term "de US outcome" mean in line 152? Spanish?

DONE. Thanks for the information. We solve the translation mistake.

  • There is a typographical error in line 175 "found that found that"

DONE. Thanks for the information. We solve the typographical error.

  • The section on results consists of only legends for Tables. The authors should move the descriptions in lines 165-185 to the Results section.

DONE. Thanks for the recommendation. We included more detail data about the results because we think that part is important to present the discussion with other authors. We also think Results section is more complete right now.

Reviewer 3 Report

The following is the summary for the present manuscript:

The aim of this study was to investigate if muscle thickness and echo intensity measured by ultrasound can discriminate post-poliomyelitis syndrome patients from healthy controls. A total of 42 postpolio patients and 39 healthy controls participated in this cross-sectional study. Anthropometric measures, muscle thickness, echo intensity using B-mode ultrasound in rectus femoris and biceps brachii muscles, and muscle strength test data were collected. Muscle thickness in rectus femoris was significantly lower in post-poliomyelitis patients than in healthy controls, but not in biceps brachii. Echo intensity in rectus femoris and biceps brachii was higher in post-poliomyelitis syndrome patients than in healthy controls. The results of the present study showed that muscle thickness in rectus femoris and echo intensity in rectus femoris and biceps brachii can discriminate post-poliomyelitis syndrome patients from healthy controls. A better assessment is possible because muscle echo intensity can ease differences between post-poliomyelitis patients and healthy controls.

I have some concerns of the present manuscript. If the authors feel it possible to resolve the following issue, I would consider to recommend its publication.

First, the introduction should be enriched by the studies investigating quantitative muscle ultrasound. The following references are suggested to be included:

https://pubmed.ncbi.nlm.nih.gov/29604401/

https://pubmed.ncbi.nlm.nih.gov/35888161/

Second, the diagnostic criteria of PPS should be given.

Third, what are their exclusion criteria of the participants?

Fourth, the experience of the examiner in musculoskeletal ultrasound shoud be detailed.

Fifth, ultrasound imaging for biceps brachii muscles should be provided.

Sixth, the muscles and adjacent structures on ultrasound images should be provided. A small side graph is needed to show the transducer placement.

Seventh, the characterics of the patients should be given. For example, which extremities are affected? How long is their disease duration?

Eighth, the annotation of the decimal point is “a point” instead a “common”.

Ninth, the correlation analysis should also be provided in graphs.

Author Response

Thank you very much for reviewing our manuscript and for providing us with detailed suggestions on how to improve it. We have addressed all the comments expressed from reviewers and hope that the explanations and revisions are satisfactory. You can find clarifications for each comment. Changes were highlighted in yellow in the manuscript.

  • First, the introduction should be enriched by the studies investigating quantitative muscle ultrasound. The following references are suggested to be included:

https://pubmed.ncbi.nlm.nih.gov/29604401/

https://pubmed.ncbi.nlm.nih.gov/35888161/

DONE. Thanks for the recommendation. We included the references in Introduction: Ultrasound variables such as muscle thickness and echo-intensity have been showed relationship with function and strength in several clinical populations [14,15].

  • Second, the diagnostic criteria of PPS should be given.

DONE. Thanks for the recommendation. We included:

The inclusion criteria were: polio survivors with new symptoms after 15 years stable; progressive appearance of muscle weakness, atrophy, pain, sensory disturbances and fatigue; persistence of new symptoms for 1 year; and no other neurological, medical or orthopedic disease that generates the symptoms.

  • Third, what are their exclusion criteria of the participants?

DONE. Thanks for the remark. We included the exclusion criteria in the Methods part:

Patients were excluded if they had a surgery intervention during the last year or any neurologic and trauma additional condition that it was impossible to carry out the tests.

  • Fourth, the experience of the examiner in musculoskeletal ultrasound shoud be detailed.

DONE. Thanks for the advice. We included the level of experience of the examiner: The examiner had more than 4 year of experience in musculoskeletal ultrasound in a clinical context.

  • Fifth, ultrasound imaging for biceps brachii muscles should be provided.

DONE. Thanks for the suggestion. We included the placement of the transductor for the biceps braquii collection in figure 1.

  • Sixth, the muscles and adjacent structures on ultrasound images should be provided. A small side graph is needed to show the transducer placement.

DONE. Thanks for the recommendation. We include a new figure with the structures in both cross-sectional images.

  • Seventh, the characterics of the patients should be given. For example, which extremities are affected? How long is their disease duration?

DONE. Thanks for the advice. We could include the affected lower limb of each patient. Unfortunately, we did not collect the disease duration time but the inclusion criteria took under consideration the importance of the time with symptoms in order to select this clinical population:

54.8% of the patients sample had the lower limb affectation in the right side, 33.3% in the left side and 11.9% in both sides.

  • Eighth, the annotation of the decimal point is “a point” instead a “common”.

DONE. As suggested, annotations of the decimal points were changed by a point. Thanks for the recommendation.

  • Ninth, the correlation analysis should also be provided in graphs.

DONE. We included in the manuscript a new figure with six graphs about the correlation analysis.

Round 2

Reviewer 2 Report

By and large, the reviewer is satisfied with the revision and would like to approve it.

Reviewer 3 Report

The article is well revised.